# Classification of *Latilactobacillus sakei* subspecies based on MALDI-TOF MS protein profiles using machine learning models

Eiseul Kim,[1] Seung-Min Yang,[1] So-Yun Lee,[1] Dae-Hyun Jung,[2] Hae-Yeong Kim[1]

**ABSTRACT** *Latilactobacillus sakei* is an important bacterial species used as a starter culture for fermented foods; however, two subspecies within this species exhibit different properties in the foods. Matrix-assisted laser desorption/ionization-time of flight mass spectrometer (MALDI-TOF MS) is the gold standard for microbial fingerprinting. However, the resolution power is down to the species level. This study was to combine MALDI-TOF mass spectra and machine learning to develop a new method to identify two *L. sakei* subspecies (*L. sakei* subsp. *sakei* and *L. sakei* subsp. *carnosus*) and non-*L. sakei* species. Totally, 227 strains were collected, with 908 spectra obtained via on- and off-plate protein extraction. Only 68.7% of strains were correctly identified at the subspecies level in the Biotyper database; however, a high level of performance was observed from the machine learning models. Partial least squares-discriminant analysis (PLS-DA), principal component analysis-K-nearest neighbor (PCA-KNN), and support vector machine (SVM) demonstrated 0.823, 0.914, and 0.903 accuracies, respectively, whereas the random forest (RF) achieved an accuracy of 0.954, with an area under the receiver operating characteristic (AUROC) curve of 0.99, outperforming the other algorithms in distinguishing the subspecies. The machine learning proved to be a promising technique for the rapid and high-resolution classification of *L. sakei* subspecies using MALDI-TOF MS.

**IMPORTANCE** *Latilactobacillus sakei* plays a significant role in the realm of food bacteria. One particular subspecies of *L. sakei* is employed as a protective agent during food fermentation, whereas another strain is responsible for food spoilage. Hence, it is crucial to precisely differentiate between the two subspecies of *L. sakei*. In this study, machine learning models based on protein mass peaks were developed for the first time to distinguish *L. sakei* subspecies. Furthermore, the efficacy of three commonly used machine learning algorithms for microbial classification was evaluated. Our results provide the foundation for future research on developing machine learning models for the classification of microbial species or subspecies. In addition, the developed model can be used in the food industry to monitor *L. sakei* subspecies in fermented foods in a time- and cost-effective method for food quality and safety.

**KEYWORDS** *Latilactobacillus sakei* subspecies, fermented food, machine learning, MALDI-TOF MS, classification

L*atilactobacillus sakei* is a bacterial species that can colonize various habitats, including sourdough, fermented cabbage, meat products, and seafood. They are considered as part of the human diet due to their presence in many foods and have been detected in human feces (1). *L. sakei* is the emblematic subspecies of foods, especially fermented sausages stored in cold temperature and vacuum-packed conditions (1). This is due to its phenotypic properties that are particularly well adapted to adverse environmental conditions such as cold, oxidation, and high salinity stress encountered during fermented food storage and processing (2). Therefore, selected strains of this species are widely

Address correspondence to Hae-Yeong Kim, hykim@khu.ac.kr.

The authors declare no conflict of interest.

See the funding table on p. 12.

used as starter cultures in industrial food fermentation (3). Additionally, *L. sakei* performs metabolic activities such as arginine degradation and purine nucleoside metabolism in food environments, producing substances that can outcompete undesired micro-organisms (4). Consequently, it is often used as a natural biopreservative in fermented foods to inhibit the growth of pathogenic and spoilage microorganisms (5). The low-temperature conditions, such as those prevalent during food product storage, are ideal for *L. sakei* (6). Therefore, it is often detected in abundance in spoiled foods, even though their role as spoilage bacteria has not been documented so far (1).

The two *L. sakei* subspecies (*L. sakei* subsp. *sakei* and *L. sakei* subsp. *carnosus*) differ based on the presence of specific soluble cell proteins (7). The ability to metabolize sugars and the acidification characteristics in meat models showed differences depending on the subspecies, possibly reflecting that one is more suited as a starter and protective culture than the other (4). The *L. sakei* subsp. *sakei* was mainly isolated from fermented fish and vegetables (8); it plays an essential role in improving the quality of fermented vegetables, such as kimchi. In addition, some of these strains are reported to have health benefits for humans. In contrast, *L. sakei* subsp. *carnosus* is mainly isolated from meat products and may play an important role in meat fermentation.

The two *L. sakei* subspecies are very closely related and share many common characteristics phylogenetically; therefore, differentiating between them can be challenging and time consuming. The identification of subspecies mainly uses molecular detection methods such as 16S rRNA gene analysis. However, the *L. sakei* subspecies cannot be distinguished by 16S rRNA gene sequencing (9). MALDI-TOF MS allows for the reliable identification of most lactic acid bacteria and is currently the main method used for identification, especially in laboratories (10). This technique offers a protein profile, considered the bacterial-specific fingerprint, allowing the high-throughput identification of large numbers of bacteria by comparison with the built-in reference spectra in the database. Nevertheless, distinguishing between these two closely related *L. sakei* subspecies remains challenging.

MALDI-TOF MS consists of complex protein profiles with numerous mass peaks that can be specified by the mass–charge ratio and the intensity (11). Appropriate use of the mass spectra can aid in identifying the bacterial species and gaining additional information from large amounts of data. Several studies have emphasized the potential of machine learning or deep learning algorithms to extract informative patterns from MALDI-TOF MS (12, 13). For example, Guerrero-López et al. developed machine learning, which facilitated the detection of carbapenemase-producing *Klebsiella pneumoniae*, using MALDI-TOF MS (14). Although many researchers have developed machine learning to classify the mass spectra of antibiotic-resistant bacteria, few studies have reported the possibility of distinguishing bacterial species or subspecies by combination of traditional MALDI-TOF MS and machine learning algorithms. The aim of this study was to evaluate whether machine learning algorithms can be used for identifying the closely related *L. sakei* subspecies using mass spectra obtained by MALDI-TOF MS.

## RESULTS

### PCR identification

Subspecies-specific PCR was performed for identifying the strains, and the PCR data were compared with MALDI-TOF MS analysis. Specific primer sets for detecting the *L. sakei* subspecies produced an amplicon for each strain. Of the 227 strains, 93 were identified as *L. sakei* subsp. *sakei* and 106 strains were identified as *L. sakei* subsp. *carnosus*. The remaining 28 strains were determined to be non-*L. sakei* species.

### Data set for classification

A total of 908 spectra were generated from 227 strains (32 reference strains and 195 isolates) using both on-plate and off-plate extractions, with two technical replicates for each method. All the spectra were divided into three classes for developing machine

learning models and were labeled as "*L. sakei* subsp. *sakei* (372 spectra)," "*L. sakei* subsp. *carnosus* (424 spectra)," and "non-*L. sakei* species (112 spectra)," based on the information about the strains. Data sets of "*L. sakei* subsp. *sakei*" and "*L. sakei* subsp. *carnosus*" were used to explore the potential for classifying the two *L. sakei* subspecies. Data sets of the "non-*L. sakei* species" and "*L. sakei* subsp. *sakei*" were used to explore the potential for identifying *L. sakei* subsp. *sakei*, whereas data sets of the "non-*L. sakei* species" and "*L. sakei* subsp. *carnosus*" were used to explore the potential for identifying *L. sakei* subsp. *carnosus*. Including the non-*L. sakei* group as a contrasting data set enhances the model's ability to distinguish between the two subspecies by providing a clear differentiation baseline. This approach allows us to identify unique patterns and features specific to each subspecies, thus improving the accuracy of the machine learning classification.

## Identification of strains using the Biotyper database

Totally, 227 strains were tested to determine whether protein profiling can be used to classify *L. sakei* subsp. *sakei*, *L. sakei* subsp. *carnosus*, and the non-*L. sakei* species in a routine MALDI Biotyper database system. Cultured single colony was subjected to MALDI-TOF analysis, and the spectra were matched by reference spectra. The identification results were calculated based on score cutoff values of spectral similarity. A score of 2.00 to 3.00 indicates high confidence in species-level identification (13). In our study, all strains belonging to the *L. sakei* subspecies were identified with average scores of 2.27 (on-plate extraction) and 2.38 (off-plate extraction), indicating a reliable species-level identification for both extraction methods. However, 61 *L. sakei* subsp. *carnosus* strains were incorrectly identified as *L. sakei* subsp. *sakei*.

## Peak analysis

Differences in the protein fingerprints between the two *L. sakei* subspecies were analyzed. The protein fingerprints of two *L. sakei* subspecies showed similar patterns (Fig. 1). Subsequently, potential subspecies-specific m/z loci were explored to observe the differences in the mass peaks between *L. sakei* subsp. *sakei* and *L. sakei* subsp. *carnosus*. A matrix was plotted based on the presence and absence of mass peaks from all strains and their degree of difference ($P < 0.01$). Several mass peaks were found to be common between the *L. sakei* subspecies; hence, despite being indicative of the species, they cannot be considered as subspecies specific. Details about the top 10 mass peak loci between the two subspecies are listed in Fig. 2. A total of 49 distinct peaks were obtained by MALDI-TOF MS. Among them, the 9,653 m/z peak was observed in 96 (90.6%) *L. sakei* subsp. *carnosus* strains and 20 (21.5%) *L. sakei* subsp. *sakei* strains (Fig. 3A). Similarly, the 4,826 m/z peak was present in 96 (90.6%) *L. sakei* subsp. *carnosus* strains and 25 (26.9%) *L. sakei* subsp. *sakei* strains (Fig. 3B). None of the mass peaks could be used to classify the two subspecies distinctly; however, they could be distinguished by a combination of mass peaks.

## Performance of machine learning models

The optimized partial least squares-discriminant analysis (PLS-DA), principal component analysis-K-nearest neighbor (PCA-KNN), support vector machine (SVM) with RBF kernel, and random forest (RF) models are known to exhibit good classification performance. The training set accuracy for PLS-DA was 0.896, and the test set accuracy was 0.823. The training set accuracy for PCA-KNN was 0.949, and the test set accuracy was 0.914. The SVM with RBF kernel showed a training set accuracy of 0.961 and a test set accuracy of 0.903, whereas the classification results in the RF were 1.000 and 0.954 for the training set and test set accuracies, respectively. Table 1 shows the evaluation performances of the four classification models, including accuracy, sensitivity, and specificity. Among the four models, RF, with the best results for accuracy (0.954), sensitivity (0.955), specificity (0.953), recall (0.955), and precision (0.955), exhibited the best classification performance. The accuracy, sensitivity, specificity, recall, and precision of the PCA-KNN model were slightly higher than those of the SVM model.

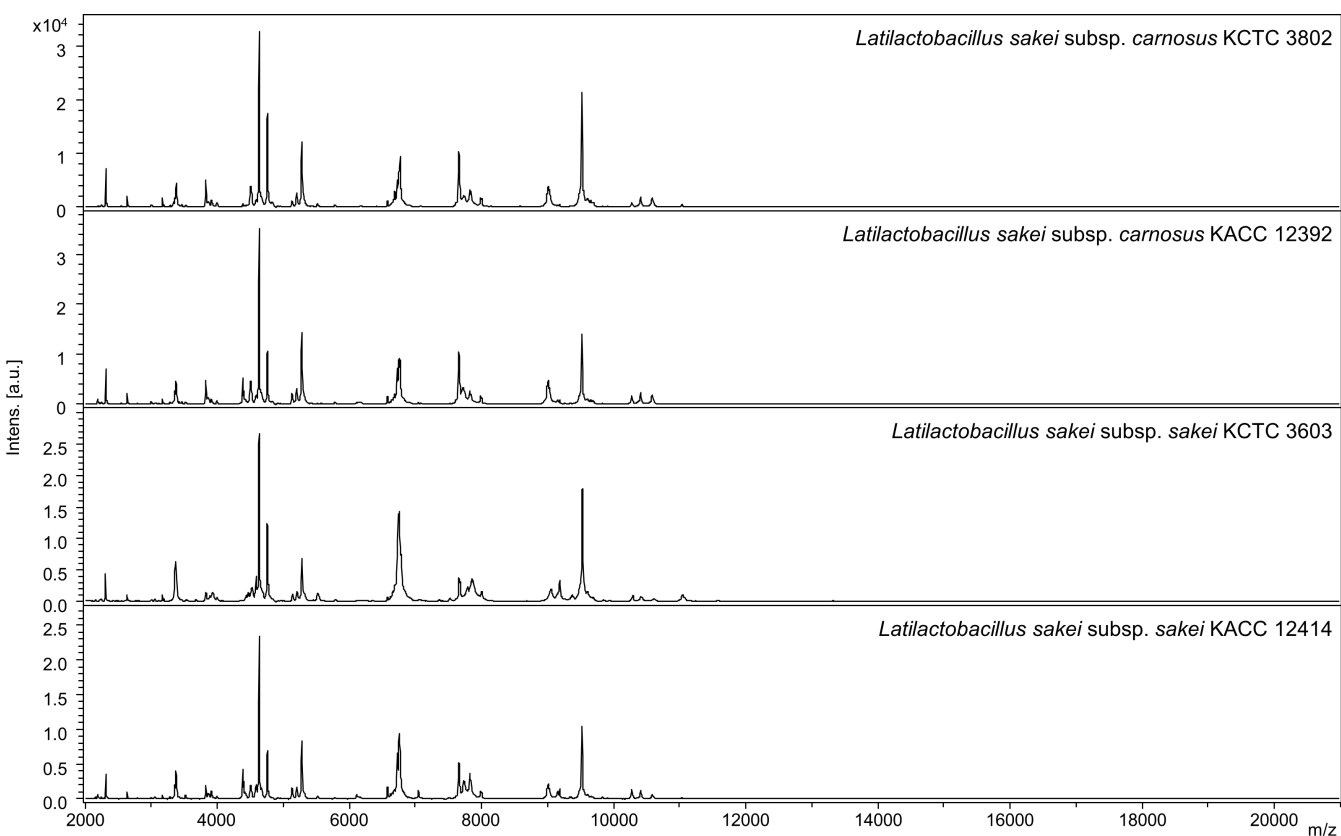

**FIG 1** Protein fingerprints of *L. sakei* subsp. *carnosus* KCTC 3802, *L. sakei* subsp. *carnosus* KACC 12392, *L. sakei* subsp. *sakei* KCTC 3603, and *L. sakei* subsp. *sakei* KACC 12414; m/z, mass-to-charge ratio; a.u., arbitrary units.

Using the same training and testing data set, the RF model misclassified eight spectra, the PCA-KNN model misclassified 15 spectra, the SVM model misclassified 17 spectra, and the PLS-DA model misclassified 31 spectra. Notably, the spectra misclassified by the RF model were also misclassified by the other models, indicating consistent misidentification across all four models. Further analysis revealed that the misclassified spectra originated from the sub-strain. However, no specific patterns or characteristics were observed that could explain the misclassification.

Additionally, the top 10% variable important peaks that determine the three classes in the RF model (Fig. 4) were identified. The top five peaks (9,653, 4,403, 4,826, 7,060, and 5,536 m/z) among the 25 peaks in the top 10% variable important features were detected. Peaks 9,653, 4,403, and 4,826 were also detected in the presence or absence matrix of mass peaks ($P < 0.01$) of two *L. sakei* subspecies by MALDIquant.

The classification performance for each model was evaluated using a confusion matrix. The numbers on the diagonal line of the matrix represented the correct number of predicted classes, whereas numbers other than zero indicated incorrect predictions. Most of the randomly selected samples from each class were satisfactorily classified (Fig. S1); however, some samples were misclassified. The PLS-DA model presented with the highest number of misclassified samples, whereas the RF model classified most of the samples satisfactorily. A more intuitive comparison of the model's performances was made using the area under the receiver operating characteristic (AUROC) curve (Fig. 5). The RF model revealed the best classification performance (AUROC = 0.99) by slightly outperforming the SVM (AUROC = 0.98) and the PCA-KNN (AUROC = 0.98) models. However, the PLS-DA model showed a lower AUROC value of 0.81.

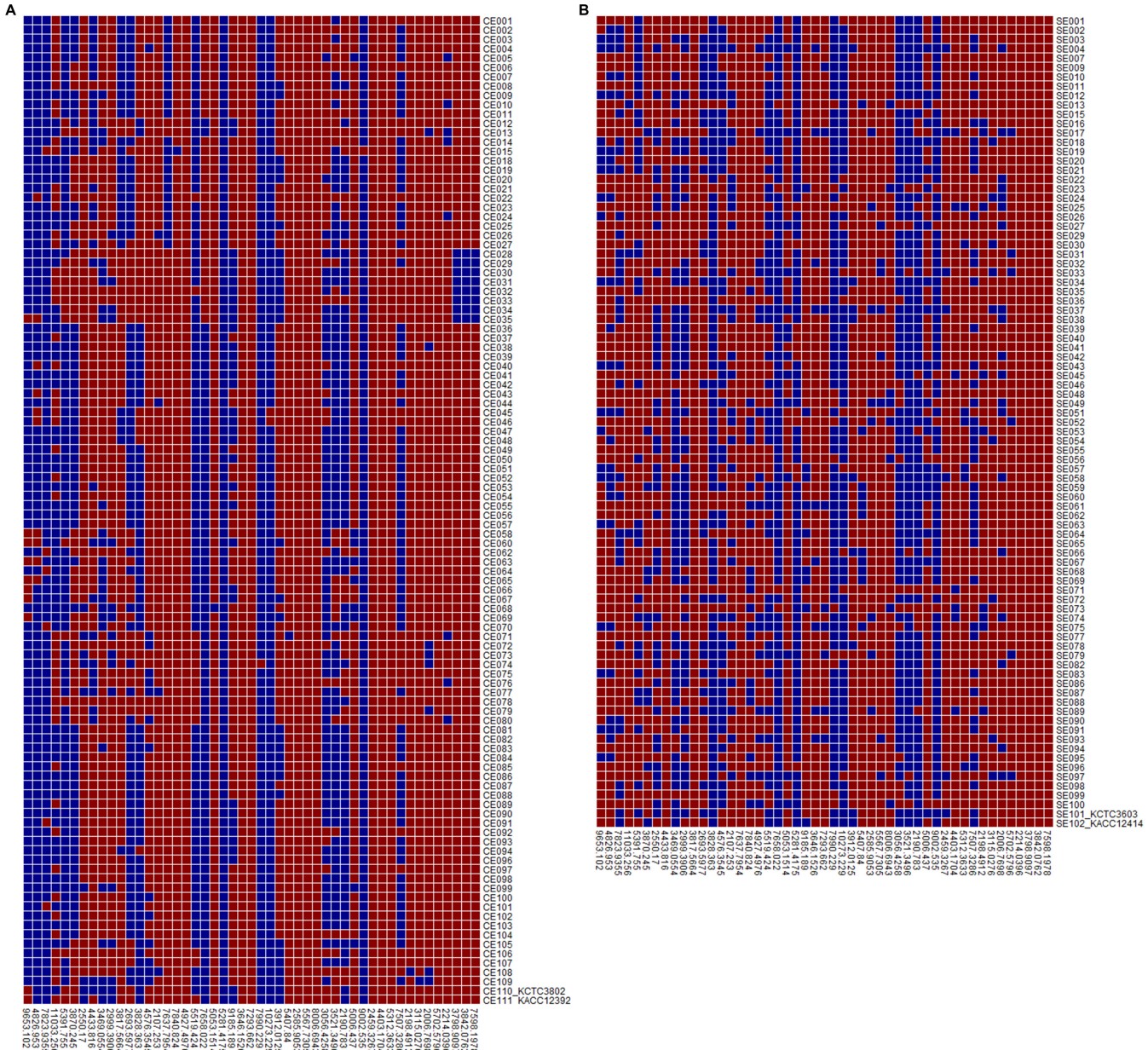

**FIG 2** The matrix with the absence and presence of mass spectral loci among *L. sakei* subsp. *sakei* and *L. sakei* subsp. *carnosus*. The mass peaks (*P* < 0.001) between *L. sakei* subsp. *sakei* and *L. sakei* subsp. *carnosus* are listed below the matrix.

## DISCUSSION

MALDI-TOF MS is widely used for identifying microorganisms owing to the following characteristics: speedy analysis, high resolving power, and relatively simple operation (15). However, this technique has some limitations with regard to meeting the full requirements for the identification of some microorganisms (16, 17). The current MALDI-TOF MS technique can accurately detect bacteria at the species level, but cannot differentiate between subspecies, strains, or drug resistance characteristics (18). Also, this technique provides low accuracy when the identifying of strains with similar protein profiling. Therefore, the techniques should be further improved by developing new artificial intelligence models or data analysis frameworks for data mining to aid in identifying bacterial subspecies (17). Several studies reported MALDI-TOF MS as a more time-effective alternative to conventional identification assays (19). The MALDI-TOF MS

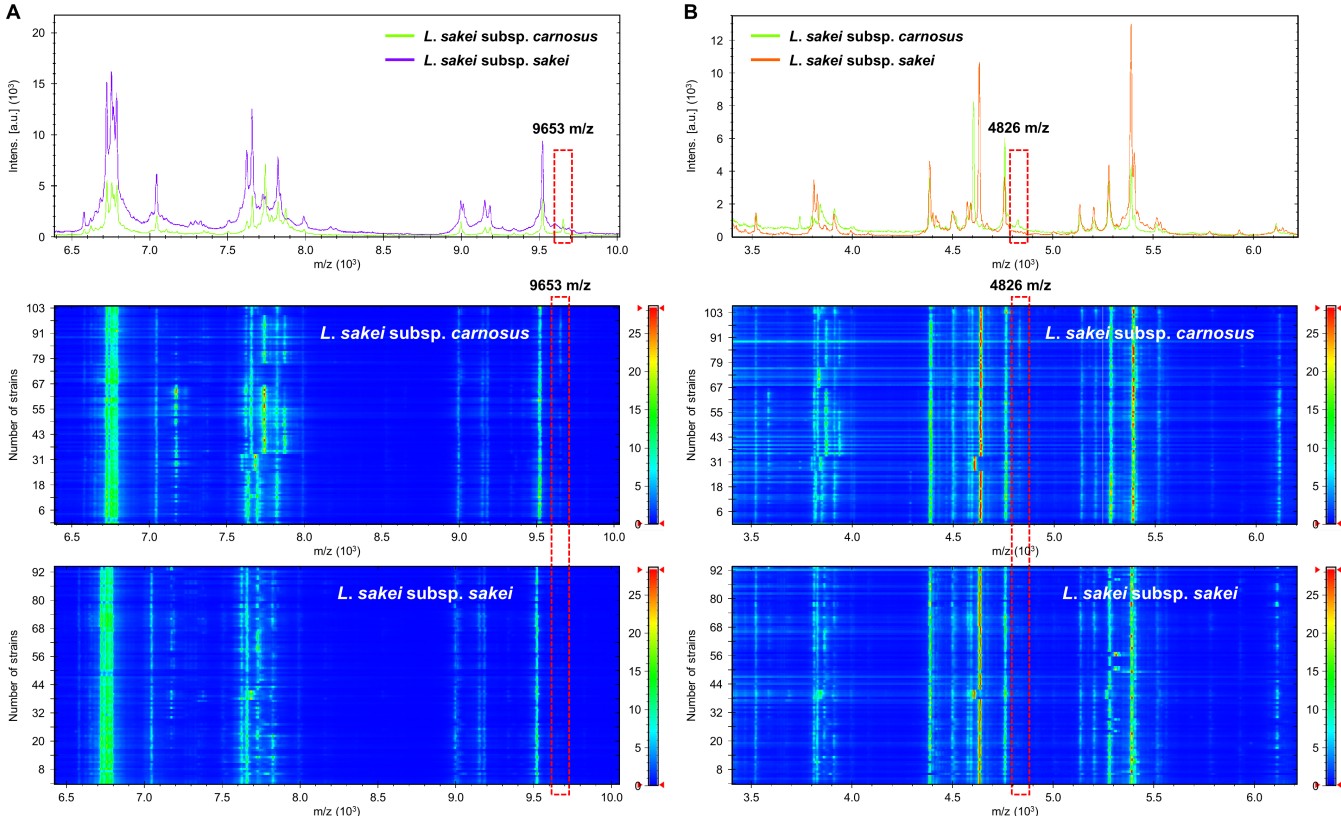

**FIG 3** Observations of the important features at (A) 9,653 and (B) 4,826 m/z for discriminating *L. sakei* subsp. *sakei* and *L. sakei* subsp. *carnosus*. In the pseudo-gel image, *x*-axis indicates m/z and *y*-axis represents the number of strains.

method, combined with machine learning, may prove as a cost-effective approach that can aid in identifying species or subspecies during routine laboratory practice (20). To the best of our knowledge, a study on the accurate discrimination of *L. sakei* subspecies using this method has not yet been published. Therefore, this study aimed to confirm whether a MALDI-TOF MS technique combined with a machine learning approach could be utilized for the rapid identification of the *L. sakei* subspecies.

 *L. sakei* is one of the most widely studied lactic acid bacteria because it inhabits various foods, resulting in the use of this species in many industrial applications. Although extensive studies have been conducted to isolate *L. sakei* from different foods, a reliable method to determine the ontogeny of this species in complex food systems is required to investigate the activity of this bacteria at the subspecies level. Additionally, the system should be advanced by a combination of data analysis algorithms to develop a MALDI-TOF MS-based bacterial species identification technique. Mass peaks can be obtained by different sample pretreatment methods or by including a large number of samples for the reproducibility and accuracy of the data (19). In the current study, the

**TABLE 1** Classification evaluation metrics for machine learning models

| Value | Machine learning models | | | |
|---|---|---|---|---|
| | PLS-DA | PCA-KNN | SVM | RF |
| Accuracy (training) | 0.896 | 0.949 | 0.961 | 1.000 |
| Accuracy (test) | 0.823 | 0.914 | 0.903 | 0.954 |
| Sensitivity | 0.928 | 0.918 | 0.917 | 0.955 |
| Specificity | 0.755 | 0.911 | 0.890 | 0.953 |
| Recall | 0.928 | 0.918 | 0.917 | 0.955 |
| Precision | 0.711 | 0.907 | 0.885 | 0.955 |

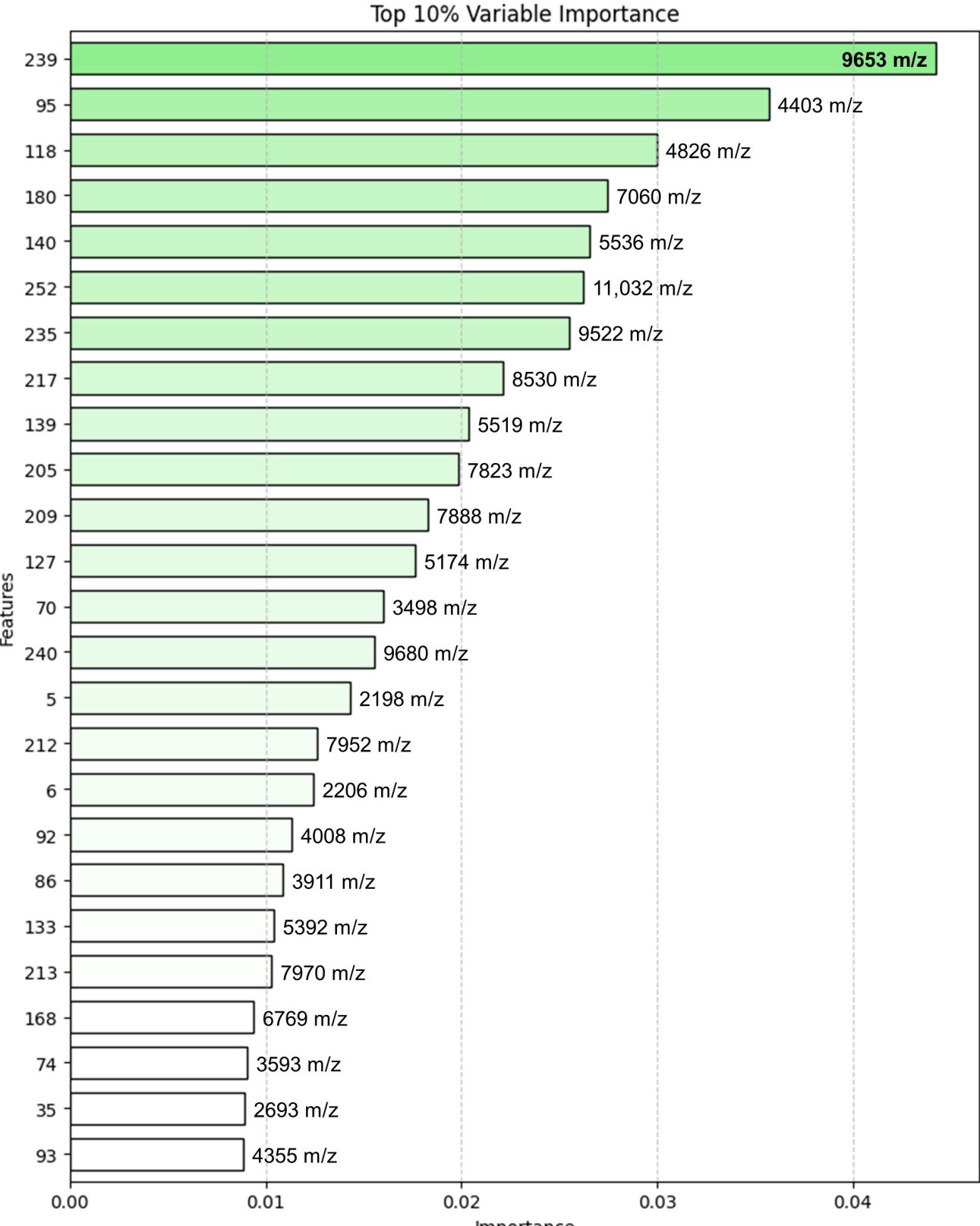

**FIG 4** Random forest top 10% variable importance plot. The variables were ranked in order of importance for classifying *L. sakei* subsp. *sakei*, *L. sakei* subsp. *carnosus*, and non-*L. sakei* species. The highest ranked features (9,653 to 4,355 m/z) contribute more to the model prediction than lower ranks and have high predictive power.

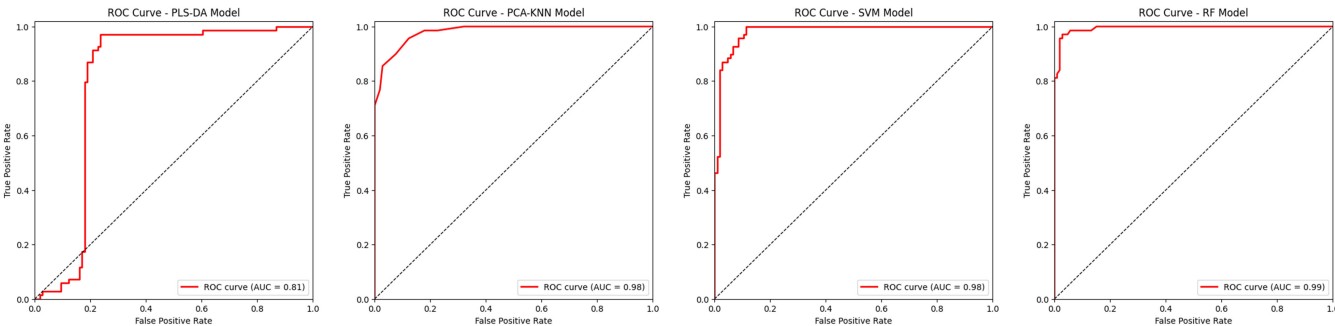

**FIG 5** AUROC curves of predicting performance of four machine learning models (PCA-KNN model, SVM model, PLS-DA model, RF model). Increasing the area under the ROC in the diagram indicates improved classification performance. The dotted lines represent the random classifier performance.

spectra of *L. sakei* subsp. *sakei*, *L. sakei* subsp. *carnosus*, and non-*L. sakei* species were obtained using different protein extraction methods. The off-plate protein extraction method is most widely used, particularly for the identification of specific mass peaks (19). However, the on-plate protein extraction method is easy to perform in routine laboratories. A machine learning model was constructed based on spectra extracted by both methods in the present study.

Currently, the Biotyper database contains one spectrum for *L. sakei* subsp. *sakei* and two spectra for *L. sakei* subsp. *carnosus*. The database was unable to distinguish between the two *L. sakei* subspecies using proteins extracted by either of the two extraction methods. However, differences were observed between the mass spectra obtained from the *L. sakei* subspecies within the mass range of 2,000–20,000 Da. Based on this observation, we combined the mass spectra with machine learning algorithms for the discrimination of *L. sakei* subspecies and non-*L. sakei* species.

In this study, we used two reference strains for each subspecies of *L. sakei*. This approach was crucial in demonstrating that our classification was not merely between two individual strains but between two distinct subspecies. By including multiple reference strains, we ensured a robust and reliable species-level identification, thereby enhancing the validity of our machine learning models and overall findings.

Advances in the machine learning algorithm have largely promoted proteomic data analysis, which makes it possible to detect the antimicrobial resistance of bacteria in a single analysis. Many researchers analyzed the antimicrobial classes, such as vancomycin and carbapenem, from their MALDI-TOF mass fingerprints (12, 21). In 2022, Weis et al. reported an antibiotic resistance prediction from the mass spectra of pathogenic bacteria via machine learning algorithm (22). In this work, machine learning algorithms, including deep neural network and logistic regression classifier, were used for model development. Yu et al. obtained accuracy values of 0.8869 and 0.8961 for predicting carbapenem-resistant and colistin-resistant *K. pneumoniae*, respectively, using the light gradient boosting machine (lightGBM) (23). Moreover, a more attractive strategy is that it can be adapted for MALDI-TOF MS-based bacterial identification and characterization (17, 24–26). In the study by Dematheis et al., machine learning models showed excellent performances with 100% accuracies (24). In another recent study, the accuracies of the SVM classifiers for classifying the important *Riemerella anatipestifer* serovar 1 and 2 and other *R. anatipestifer* serovars were 76.27% and 83.00%, respectively, and those of the RF classifiers were 76.34% and 83.33%, respectively (27). Interestingly, they discovered the 2,530 m/z peak for classifying the two *R. anatipestifer* serovars. In the current study, the accuracy described by the previously mentioned studies was lower than the species classification and higher than the serovar classification model (accuracy = 0.954).

In this study, machine learning models were developed based on four algorithms (PLS-DA, PCA-KNN, SVM, and RF), with the model based on the RF algorithm showing the best performance. The accuracies of the machine learning algorithm are mainly because of its mathematical properties (28). The RF algorithm can construct numerous

decision trees classified by the number of data independently of each other, ensuring the robustness of the algorithm. The RF algorithm can process high-dimensional variable data sets despite the presence of irrelevant variables. In a previous study, five algorithms were compared to classify the *Mycobacterium* subspecies, and the model constructed by the RF algorithm demonstrated the best performance (accuracy = 0.9166) (29). In addition, the top informative peaks (6,715 and 4,739 m/z) were identified. Consistent with these findings, the performance of the RF algorithm was excellent in the current study, and the accuracy, sensitivity, specificity, recall, and precision values were >0.95. Additionally, a high AUROC value (0.99) was observed with the RF model for classifying *L. sakei* subspecies and non-*L. sakei* species, indicating that the current model may accurately identify the *L. sakei* subspecies. Furthermore, informative peaks (9,653 and 4,826 m/z) were identified by the RF method in the present study. These putative markers were discovered in three classes of the *L. sakei* subsp. *sakei*, *L. sakei* subsp. *carnosus*, and non-*L. sakei* species by investigating the RF algorithm features of importance. Nevertheless, interpretations of the biological roles of the features remain elusive. To determine the biological functions of these markers, peptide sequencing should be performed (30, 31).

In conclusion, this study combined mass spectra obtained from MALDI-TOF MS and machine learning to classify the *L. sakei* subspecies, which play an important role in food fermentation. The results showed that the model based on the RF algorithm had the best accuracy for *L. sakei* subspecies classification. The developed machine learning model allows for the identification and monitoring of specific subspecies rapidly and accurately. Thus, this method can be used for quality control food quality in the fermented food industry.

## MATERIALS AND METHODS

### Isolation of *L. sakei*

Thirty-one samples of fermented fish (*n* = 11), fermented vegetables (*n* = 9), and fermented meat (*n* = 11) were collected from supermarkets in Korea (Table S1). The *L. sakei* subspecies were isolated by homogenizing 10 g of food sample with 90 mL of phosphate-buffered saline for 1 min. Serial dilutions were plated on de Man-Rogosa-Sharpe (MRS) agar (Difco, Franklin Lakes, NJ, USA) at 30°C. After 48 h of inoculation, each colony was transferred into MRS broth and was further cultured at 30°C for 48 h. The 32 reference strains obtained from the Korean Agricultural Culture Collection (KACC, Wanju, Korea), Korean Collection for Type Culture (KCTC, Daejeon, Korea), and Korean Culture Center of Microorganisms (KCCM, Seoul, Korea) were used for the MALDI-TOF MS analysis (Table 2).

### Molecular identification by subspecies-specific PCR

Isolates from food samples were identified by colony PCR with specific primers (Table 3) reported in our previous study (9). For DNA extraction, isolates were sub-cultured in MRS agar at 30°C. A single colony was dissolved in 50 µL of distilled water and heated at 95°C for 5 min. After centrifugation, the supernatant was used as a template. PCR amplification of subspecies-specific genetic markers was performed using the specific primers and the conditions described in our previous study (9).

### Protein extraction

The proteins of strains for the MALDI-TOF MS were extracted using the on-plate and off-plate protein extraction methods. In the on-plate protein extraction method, a single colony was smeared on a plate (Bruker Daltonik GmbH), covered with 1 µL of 70% formic acid and matrix solution (Bruker Daltonik GmbH), which consisted of a saturated solution. In the off-plate protein extraction method, 200 mg of the bacterial cells were transferred to an extraction tube containing 300 µL of pure water. After adding ethanol (0.9 mL), the tubes were centrifuged at 13,000 × *g* for 2 min; the supernatants were

**TABLE 2**  Bacterial strains used in this study

| Species or subspecies | Origins |
|---|---|
| Reference strains | |
| *Latilactobacillus sakei* subsp. *carnosus* KACC 12392 | Fermented meat product |
| *Latilactobacillus sakei* subsp. *carnosus* KCTC 3802 | Fermented meat product |
| *Latilactobacillus sakei* subsp. *sakei* KACC 12414 | Moto starter of sake |
| *Latilactobacillus sakei* subsp. *sakei* KCTC 3603 | Moto starter of sake |
| *Latilactobacillus curvatus* KACC 12415 | Milk |
| *Latilactobacillus curvatus* KCTC 3767 | Milk |
| *Latilactobacillus fuchuensis* KCTC 3797 | Beef |
| *Apilactobacillus kunkeei* KACC 19371 | Commercial grape wine |
| *Companilactobacillus farciminis* KCTC 3681 | Sausage |
| *Companilactobacillus heilongjiangensis* KACC 18741 | Traditional pickle |
| *Lactiplantibacillus paraplantarum* KCTC 5045 | Beer contaminant |
| *Lactiplantibacillus plantarum* KCTC 3104 | Pickled cabbage |
| *Lactobacillus acidophilus* KCTC 3164 | Human feces |
| *Lactobacillus amylovorous* KCTC 3597 | Cattle waste-corn fermentation |
| *Lactobacillus crispatus* KACC 12439 | Eye |
| *Lactobacillus delbrueckii* KACC 12420 | Bulgarian yogurt |
| *Lactobacillus delbrueckii* KACC 13439 | Sour grain mash |
| *Lactobacillus delbrueckii* KCTC 21031 | Malted sorghum |
| *Lactobacillus delbrueckii* KACC 12417 | Cheese |
| *Lactobacillus delbrueckii* KCTC 15515 | Fermented vegetable |
| *Lactobacillus gallinarum* KACC 12370 | Chicken crop |
| *Lactobacillus gasseri* KACC 12424 | Human |
| *Lactobacillus graminis* KCTC 3542 | Grass silage |
| *Lactobacillus helveticus* KACC 12418 | Cheese |
| *Lactobacillus johnsonii* KCTC 3801 | Human blood |
| *Lactobacillus malefermentans* KCCM 40873 | Beer |
| *Lactobacillus pentosus* KCCM 40997 | Corn silage |
| *Levilactobacillus brevis* KCTC 3498 | Human feces |
| *Levilactobacillus zymae* KACC 16349 | Artisanal wheat sourdough |
| *Ligilactobacillus acidipiscis* KACC 12394 | Fermented fish |
| *Limosilactobacillus fermentum* KCTC 3112 | Fermented beets |
| *Limosilactobacillus reuteri* subsp. *reuteri* KCTC 3594 | Intestine of adult |
| [a]Isolates (number of strains) | |
| *L. sakei* subsp. *sakei* (91) | Fermented fishes and vegetables |
| *L. sakei* subsp. *carnosus* (104) | Fermented fishes and meat |

[a]Detailed information is shown in Table S1.

discarded, and the pellets were air dried. The pellet was thoroughly mixed with 20 µL of formic acid and acetonitrile. The supernatant was spotted on MALDI plate and overlaid with 1 µL of the matrix solution.

## Acquisition of the MALDI-TOF MS spectra

Measurements were performed using a microflex LT MALDI-TOF MS (Bruker Daltonik GmbH) and the FlexControl software (version 3.1). The protein mass fingerprints were

**TABLE 3**  Subspecies-specific primers used in this study

| Target | Sequence (5′–3′) | Size (bp) | Reference |
|---|---|---|---|
| *L. sakei* subsp. *sakei* | CGC GAT TGA TGA CGG TAT TC | 116 | (9) |
| | GCG TGC CAT CAT CAT TAC CA | | |
| *L. sakei* subsp. *carnosus* | ACT TAC CGA TCC TGG CTG TC | 157 | (9) |
| | ACT TAC CGA TCC TGG CTG TC | | |

obtained in the linear positive mode (intensity 95%; ion source one = 18.00 kV; ion source two = 16.38 kV; lens = 5.40 kV; pulse ion extraction = 150 ns) at a laser frequency of 60 Hz within a mass range of 2,000 to 20,000 Da. A minimum of 40 laser shots per sample was used to generate each ion spectrum. Calibration was performed using the Bruker Bacterial Test Standard (Bruker Daltonik GmbH).

## Data preprocessing

Data preprocessing and quality control of MALDI-TOF mass spectra were conducted using R, version 4.3.1. MALDIquant package was used to preprocess the mass spectra. First, a square root transformation was applied, followed by baseline correction and intensity normalization using the statistics-sensitive nonlinear iterative peak-clipping method and the total ion current method, respectively. Next, peak detection was performed using the peak binning function. Finally, the feature matrix containing the peak intensity information was generated using the intensityMatrix function.

## Machine learning

After preprocessing the mass spectra data, four machine learning algorithms, including PLS-DA, PCA-KNN, SVM with RBF kernel, and RF, were employed to construct the classification models for *L. sakei* subsp. *sakei*, *L. sakei* subsp. *carnosus*, and the non-*L. sakei* species. The four machine learning algorithms were run on Python version 3.8, and the training and testing data sets were randomly split. Specifically, 80% of the samples were used as a training data set, whereas the remaining 20% were used as the test data set. All models were implemented with their default parameter settings. The PCA-KNN algorithm used principal components that explained 95% of the variance and a k value of 9 for the KNN classifier. The SVM with RBF kernel had a C value of 1.0 and a gamma value of "scale." For the RF algorithm, the parameters included 100 estimators, a minimum sample split of 2, and "auto" for maximum features. Four possible outcomes for the machine learning models were predicted: true positive (TP), true negative (TN), false positive (FP), and false negative (FN) (28). The classification performance was evaluated using the performance metrics precision, accuracy, sensitivity, specificity, and recall, as defined in the following formulas:

$$Precision = \frac{TP}{TP + FP} \tag{1}$$

$$Accuracy = \frac{TP + TN}{TP + TN + FP + FN} \tag{2}$$

$$Sensitivity = \frac{TP}{TP + FN} \tag{3}$$

$$Specificity = \frac{TN}{TN + FP} \tag{4}$$

$$Recall = \frac{TP}{TP + FN} \tag{5}$$

Precision refers to the ratio of the positive results to overall predicted results (28). Accuracy refers to the decision ability of the classifier for all the samples. Specificity is a measure of how well a model can detect TN samples, whereas sensitivity is a measure of how well a model can detect TP samples. Recall is the ratio of the correctly predicted positive results out of all the positive samples. The AUROC curve was generated for evaluating the models.

## ACKNOWLEDGMENTS

This work was supported by the National Research Foundation of Korea (NRF), grant funded by the Korea government (MSIT) (No. RS-2023-00212751).

## AUTHOR AFFILIATIONS

[1]Department of Food Science and Biotechnology, Institute of Life Sciences & Resources, Kyung Hee University, Yongin, South Korea

[2]Department of Smart Farm Science, Kyung Hee University, Yongin, South Korea

## AUTHOR ORCIDs

Eiseul Kim ⓘ http://orcid.org/0000-0003-0969-4048

Hae-Yeong Kim ⓘ http://orcid.org/0000-0003-3409-0932

## FUNDING

| Funder | Grant(s) | Author(s) |
|---|---|---|
| National Research Foundation of Korea (NRF) | No. RS-2023-00212751 | Eiseul Kim |

## AUTHOR CONTRIBUTIONS

Eiseul Kim, Conceptualization, Data curation, Investigation, Methodology, Writing – original draft | Seung-Min Yang, Data curation, Investigation | So-Yun Lee, Investigation | Dae-Hyun Jung, Writing – review and editing | Hae-Yeong Kim, Conceptualization, Resources, Supervision, Writing – review and editing

## DATA AVAILABILITY

MALDI-TOF MS spectra are available in a data set on Dryad. The data set can be accessed via the following DOI: https://doi.org/10.5061/dryad.ncjsxkt4q.

## ADDITIONAL FILES

The following material is available online.

### Supplemental Material

**Supplemental material (Spectrum03668-23-s0001.docx).** Fig. S1; Table S1.

### Open Peer Review

**PEER REVIEW HISTORY (review-history.pdf).** An accounting of the reviewer comments and feedback.

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
