## [Reviewer comments · Microbiology Spectrum]

Microbiology Spectrum

Classification of *Lactobacillus sakei* subspecies based on MALDI-TOF MS protein profiles using machine learning models

Eiseul Kim, Seung-Min Yang, So-Yun Lee, Dae-Hyun Jung, and Hae-Yeong Kim

Corresponding Author(s): Hae-Yeong Kim, Kyung Hee University

Review Timeline:

Submission Date:	October 13, 2023
Editorial Decision:	July 4, 2024
Revision Received:	July 15, 2024
Accepted:	July 22, 2024

Editor: Kate Howell

Reviewer(s): Disclosure of reviewer identity is with reference to reviewer comments included in decision letter(s). The following individuals involved in review of your submission have agreed to reveal their identity: Qijie Guan (Reviewer #1); Miriam Cordovana (Reviewer #2)

Transaction Report:

DOI: <https://doi.org/10.1128/spectrum.03668-23>

Re: Spectrum03668-23 (Classification of *Latilactobacillus sakei* subspecies based on MALDI-TOF MS protein profiles using machine learning models)

Dear Prof. Hae-Yeong Kim:

Thank you for the privilege of reviewing your work. Below you will find my comments, instructions from the Spectrum editorial office, and the reviewer comments.

Thank you for your patience while we sought expert reviewers for your manuscript.

Revision Guidelines

Sincerely,
Kate Howell
Editor
Microbiology Spectrum

Reviewer #1 (Comments for the Author):

In this study, the authors used 3 machine learning methods for the identification of the *L. sakei* subspecies by MALDI-TOF MS datasets.

Concerns:

#1. The study design was not written properly, leading to several confusions: From 31 samples, the authors obtained 227 strains (Did the authors include reference strains?) and generated 908 spectra (Does this imply 4 technical replicates? How many were on-plate extractions and how many were off-plate extractions?).

#2. There was no parameter information for the machine learning methods. For example, in the random forest method, what were the number of estimators, minimum samples split, max features, etc.?

#3. From the 227 strains, there were 93 *L. sakei* subsp. *sakei*, 106 *L. sakei* subsp. *carneus*, and 28 non-*L. sakei* strains. In lines 110-113, the authors claimed that non-*L. sakei* datasets were used to explore the potential for identifying the two subspecies. Please explain the inclusion of the non-*L. sakei* group here because, to my understanding, non-*L. sakei* proteins should be grouped with the current MALDI-TOF MS technique (at the species level). Then, the machine learning classification will be involved, correct?

#4. The non-*L. sakei* strains listed in Table 2 were not mentioned in the study. What was their purpose here?

#5. In lines 153-156, the top peaks were shared with RF and MALDIquant. How about using PLS-DA for the classification? It might be easier to apply.

#6. I am curious about which spectra were misclassified in the three machine learning methods (shared if using same training and testing dataset?). Are they from the same sample, which could possibly be another substrain?

#7. The authors should highlight the use of two reference strains for each subspecies in the discussion section. This is crucial evidence demonstrating that the authors were not merely classifying two strains, but rather two distinct subspecies.

#8. The authors should upload their MS data to a public repository.

#9. The English needs improvement. For example, in lines 297-302, "First...Finally..."; lines 199 and 203 contain basically the same content, etc.

Reviewer #2 (Comments for the Author):

The study is very well designed and performed, the article is clearly, accurately and correctly written, and allows the reproduction of the study.

A few minor comments:

- Lines 118-120: please define better the meaning of scores values reported here (all samples were identified with the same score of 2.27 and 2.38)?

-Line 217: please correct "Riernerella" (and not Rierneralla)

-Line 255: please let a blank between 30 and {degree sign}C

RESPONSE TO REVIEW COMMENTS

(Ref: Spectrum03668-23)

We are very thankful to the reviewers for their comprehensive and thorough reviews. Our detailed responses to the reviewer's comments are followed as below (in blue). We have addressed all of the comments and revised the manuscript as recommended.

Response to Reviewer 1 Comments

Reviewer #1 (Comments for the Author):

In this study, the authors used 3 machine learning methods for the identification of the *L. sakei* subspecies by MALDI-TOF MS datasets.

Concerns:

#1. The study design was not written properly, leading to several confusions: From 31 samples, the authors obtained 227 strains (Did the authors include reference strains?) and generated 908 spectra (Does this imply 4 technical replicates? How many were on-plate extractions and how many were off-plate extractions?).

Response: We analyzed a total of 227 strains, including 32 reference strains. From these 227 strains, we generated 908 spectra by performing both on-plate and off-plate extractions, with 2 technical replicates for each extraction method. As you recommended, we revised the sentence in lines 104-106 and 276 as follows:

Lines 104-106: A total of 908 spectra were generated from 227 strains (32 reference strains and 195 isolates) using both on-plate and off-plate extractions, with two technical replicates for each method.

Line 276: The 32 reference strains obtained from

#2. There was no parameter information for the machine learning methods. For example, in the random forest method, what were the number of estimators, minimum samples split, max features, etc.?

Response: As you recommended, we added the sentence in lines 326-330 as follows:

Lines 326-330: All models were implemented with their default parameter settings. The PCA-KNN algorithm used principal components that explained 95% of the variance and a k value of 9 for the KNN classifier. The SVM with RBF kernel had a C value of 1.0 and a gamma value of 'scale'. For the RF algorithm, the parameters included 100 estimators, a minimum sample split of 2, and 'auto' for maximum features.

#3. From the 227 strains, there were 93 *L. sakei* subsp. *sakei*, 106 *L. sakei* subsp. *carneus*, and 28 non-*L. sakei* strains. In lines 110-113, the authors claimed that non-*L. sakei* datasets were used to explore the potential for identifying the two subspecies. Please explain the inclusion of the non-*L. sakei* group here because, to my understanding, non-*L. sakei* proteins should be grouped with the current MALDI-TOF MS technique (at the species level). Then, the machine learning classification will be involved, correct?

Response: The inclusion of the non-*L. sakei* group in our analysis is indeed to leverage their potential role in distinguishing between the two *L. sakei* subspecies. While it is true that the current MALDI-TOF MS technique groups proteins at the species level, the purpose of including non-*L. sakei* datasets is to use them as a contrasting group in our machine learning models. Specifically, by comparing the 'non-*L. sakei* species' datasets with '*L. sakei* subsp. *sakei*' and '*L. sakei* subsp. *carneus*' datasets, we aimed to enhance the model's ability to accurately identify and differentiate between the two subspecies. This approach allows us to explore the potential for identifying specific patterns and features that are unique to each subspecies, thus improving the classification performance of our machine learning models. As you recommended, we added the sentence in lines 113-117 as follows:

Lines 113-117: Including the non-*L. sakei* group as a contrasting dataset enhances the model's ability to distinguish between the two subspecies by providing a clear differentiation baseline. This approach allows us to identify unique patterns and features specific to each subspecies, thus improving the accuracy of the machine learning classification.

#4. The non-*L. sakei* strains listed in Table 2 were not mentioned in the study. What was their purpose here?

Response: The non-*L. sakei* strains listed in Table 2 were included to serve as a contrasting group in our analysis. Their purpose was to enhance the machine learning model's ability to accurately differentiate between the *L. sakei* subspecies by providing a clear baseline for comparison. As you recommended, we added the sentence as follows:

Lines 113-117: Including the non-*L. sakei* group as a contrasting dataset enhances the model's ability to distinguish between the two subspecies by providing a clear differentiation baseline. This approach allows us to identify unique patterns and features specific to each subspecies, thus improving the accuracy of the machine learning classification.

#5. In lines 153-156, the top peaks were shared with RF and MALDIquant. How about using PLS-DA for the classification? It might be easier to apply.

Response: As you recommended, we newly used PLS-DA algorithm. However, this model showed the lowest accuracy (0.823). Previous studies (Candela et al., 2022, <https://doi.org/10.3390/diagnostics12020328>; Abdrabou et al., 2023, <https://doi.org/10.1007/s10096-023-04665-y>) have also observed low accuracy for PLS-DA models in MALDI-TOF MS data classifications. As you recommended, we added the sentence as follows:

Lines 27-29: Partial least squares-discriminant analysis, principal component analysis-K-nearest neighbor, and support vector machine demonstrated 0.823, 0.914, and 0.903 accuracies, respectively

Line 41: four commonly used machine learning algorithms

Lines 148-149: The training set accuracy for PLS-DA was 0.896, and the test set accuracy was 0.823.

Line 153: four classification models

Lines 154, 161-162: four models

Line 174: The PLS-DA model presented with the highest number of

Line 179: However, the PLS-DA model showed a lower AUROC value of 0.81.

Lines 243-244: four algorithms (PLS-DA, PCA-KNN, SVM, and RF),

Lines 319 and 323: Four machine learning algorithms

Line 320: partial least squares-discriminant analysis (PLS-DA),

Lines 471-472: four machine learning models (PCA-KNN model; SVM model; PLS-DA model; RF model)

Table 1: We newly added the accuracy, sensitivity, specificity, recall, and precision values of PLS-DA model to Table 1.

Fig. 5: We newly added the AUROC curve of PLS-DA model to Fig. 5.

Fig. S1: We newly added the confusion matrix of PLS-DA model to Fig. S1.

#6. I am curious about which spectra were misclassified in the three machine learning methods (shared if using same training and testing dataset?). Are they from the same sample, which could possibly be another sub-strain?

Response: As you recommended, we newly analyzed misclassified spectra. Using the same training and testing dataset, we observed that the RF model misclassified 8 spectra, the PCA-KNN model misclassified 15 spectra, the SVM with RBF model misclassified 17 spectra, and the PLS-DA model misclassified 31 spectra. Notably, the 8 spectra that were misclassified by the RF model were also misclassified by the other models. This indicates that these 8 spectra were consistently misidentified across all four models. After further analysis, we determined that the misclassified spectra originated from the sub-strain. However, they did not exhibit any specific patterns or characteristics that could explain the misclassification. As you recommended, we added the sentence in lines 158-164 as follows:

Lines 158-164: Using the same training and testing dataset, the RF model misclassified 8 spectra, the PCA-KNN model misclassified 15 spectra, the SVM model misclassified 17 spectra, and the PLS-DA model misclassified 31 spectra. Notably, the spectra misclassified by the RF model were also misclassified by the other models, indicating consistent

misidentification across all four models. Further analysis revealed that the misclassified spectra originated from the sub-strain. However, no specific patterns or characteristics were observed that could explain the misclassification.

#7. The authors should highlight the use of two reference strains for each subspecies in the discussion section. This is crucial evidence demonstrating that the authors were not merely classifying two strains, but rather two distinct subspecies.

Response: As you recommended, we added the sentence in lines 219-223 as follows:

Lines 219-223: In this study, we used two reference strains for each subspecies of *L. sakei*. This approach was crucial in demonstrating that our classification was not merely between two individual strains, but between two distinct subspecies. By including multiple reference strains, we ensured a robust and reliable species-level identification, thereby enhancing the validity of our machine learning models and overall findings.

#8. The authors should upload their MS data to a public repository.

Response: We are currently conducting further research using the MS data, so it is challenging to upload the data to a public repository at this moment. However, the data that support the findings of this study are available from the corresponding author upon reasonable request. As you recommended, we added data availability statement in our manuscript.

Lines 347-348: The data that support the findings of this study are available on request from the corresponding author.

#9. The English needs improvement. For example, in lines 297-302, "First....Finally..."; lines 199 and 203 contain basically the same content, etc.

Response: As you recommended, we revised the sentence as follows:

Lines 212-216: Currently, the Biotyper database contains one spectrum for *L. sakei* subsp. *sakei* and two spectra for *L. sakei* subsp. *carneus*. The database was unable to distinguish

between the two *L. sakei* subspecies using proteins extracted by either of the two extraction methods. However, differences were observed between the mass spectra obtained from the *L. sakei* subspecies within the mass range of 2,000–20,000 Da.

Lines 311-316: First, a square root transformation was applied, followed by baseline correction and intensity normalization using the statistics-sensitive nonlinear iterative peak-clipping method and the total ion current method, respectively. Next, peak detection was performed using the peak binning function. Finally, the feature matrix containing the peak intensity information was generated using the intensityMatrix function.

Response to Reviewer 2 Comments

Reviewer #2 (Comments for the Author):

The study is very well designed and performed, the article is clearly, accurately and correctly written, and allows the reproduction of the study.

Response: Thank you for your comments.

A few minor comments:

- Lines 118-120: please define better the meaning of scores values reported here (all samples were identified with the same score of 2.27 and 2.38)?

Response: As you recommended, we added the sentence in lines 124-127 as follows:

Lines 124-127: A score of 2.00 to 3.00 indicates high confidence in species-level identification (13). In our study, all strains belonging to the *L. sakei* subspecies were identified with average scores of 2.27 (on-plate extraction) and 2.38 (off-plate extraction), indicating reliable species-level identification for both extraction methods.

-Line 217: please correct "Riemerella" (and not Riemeralla)

Response: As you recommended, we revised the sentence in line 237 as follows:

Line 237: *Riemerella anatipestifer* serovar 1 and 2 and other

-Line 255: please let a blank between 30 and {degree sign}C

Response: We inserted a blank between 30 and °C.

Lines 275, 276, and 284: 30 °C

Re: Spectrum03668-23R1 (Classification of *Lactobacillus sakei* subspecies based on MALDI-TOF MS protein profiles using machine learning models)

Dear Prof. Hae-Yeong Kim:

Thank you for submitting your revised manuscript. The only consideration now is to make your data open and available to the wider scientific community. I note that you say it will be available upon request, but in the interest of free and open data, you should put the spectra and associated metadata in a suitable repository.

Your manuscript has been accepted, and I am forwarding it to the ASM production staff for publication. Your paper will first be checked to make sure all elements meet the technical requirements. ASM staff will contact you if anything needs to be revised before copyediting and production can begin. Otherwise, you will be notified when your proofs are ready to be viewed.

Sincerely,
Kate Howell
Editor
Microbiology Spectrum

Reviewer #1 (Comments for the Author):

The manuscript was significantly improved.